# Oxytetracycline Persistence in Uterine Secretion after Intrauterine Administration in Cows with Metritis

**DOI:** 10.3390/ani12151922

**Published:** 2022-07-28

**Authors:** Rositsa Mileva, Manol Karadaev, Ivan Fasulkov, Nikolina Rusenova, Nasko Vasilev, Aneliya Milanova

**Affiliations:** 1Department of Pharmacology, Animal Physiology, Biochemistry and Chemistry, Faculty of Veterinary Medicine, Trakia University, 6000 Stara Zagora, Bulgaria; rositsamileva88@gmail.com; 2Department of Obstetrics, Reproduction and Reproductive Disorders, Faculty of Veterinary Medicine, Trakia University, 6000 Stara Zagora, Bulgaria; karadaev@abv.bg (M.K.); i.fasulkov@gmail.com (I.F.); nasvas@abv.bg (N.V.); 3Department of Veterinary Microbiology, Infectious and Parasitic Diseases, Faculty of Veterinary Medicine, Trakia University, 6000 Stara Zagora, Bulgaria; n_v_n_v@abv.bg

**Keywords:** cows, intrauterine disposition, metritis, oxytetracycline

## Abstract

**Simple Summary:**

Puerperal metritis in cows involves the acute inflammation of the uterus, which is often treated with antibacterial drugs. Restrictions on the use of antibiotics in veterinary medicine require the very precise selection of an antibiotic based on its pharmacokinetics and on sensitivity tests of pathogenic bacteria. This study aimed to evaluate the changes in oxytetracycline levels in uterine secretion over time after intrauterine administration in cows with metritis in relation to the sensitivity of pathogenic bacterial isolates. The concentrations of oxytetracycline in the uterine secretions were higher than the minimum inhibitory concentrations against pathogenic bacteria, provoking the infection of the uterus. Higher concentrations were measured in milk rather than in plasma. The intrauterine infusion of oxytetracycline for three consecutive days led to the alleviation of the inflammation and the restoration of the estrus cycle in cows. The local intrauterine administration of oxytetracycline requires the monitoring of the withdrawal time of milk to ensure consumer safety.

**Abstract:**

Puerperal metritis in cows is often treated with antibacterial drugs. The prudent use of antibiotics in farm animals requires state-of-the-art knowledge of their pharmacokinetics and data from sensitivity tests of pathogenic bacteria. Changes in oxytetracycline levels in the uterine secretion over time after intrauterine administration in cows with metritis were evaluated in relation to the sensitivity of pathogenic bacterial isolates. Oxytetracycline levels in plasma, milk and uterine secretion were determined via LC–MS/MS analysis. Pathogenic bacteria were isolated and their sensitivity to oxytetracycline was determined. The concentrations of oxytetracycline in the uterine secretion were 433.79 (39.17–1668.76) µg·mL^−1^ six hours after the third application at a dose of 8 mg/kg and 84.33 (1.58–467.55) µg·mL^−1^ 96 h after the last treatment. These levels were higher than the minimum inhibitory concentrations—namely, between 4 and 64 µg·mL^−1^—against pathogenic isolates *Trueperella pyogenes*, *Streptococcus intermedius*, *Escherichia coli* and *Bacillus pumilis*. Higher concentrations over time were measured in milk rather than in plasma, indicating the need to monitor the withdrawal time for milk. The intrauterine infusion of oxytetracycline for three consecutive days resulted in the successful treatment of metritis in terms alleviating inflammation and restoring the estrus cycle in cows.

## 1. Introduction

Metritis is a postpartum disease that can cause reproductive inefficiency in dairy cattle [1]. The disease is associated with bacterial intrauterine contamination by aerobic and anaerobic Gram-positive and Gram-negative microorganisms [2]. Several studies have revealed *Trueperella pyogenes* as one of the main causative agents [3]. Usually, metritis is diagnosed during the first 21 days after parturition and can be treated by the local application of antibiotics or by the systemic administration of antibacterial drugs in cases where cows become febrile [4]. The published literature shows no uniform approach in the treatment of the disease and controversial reports regarding the efficacy of systemic and local intrauterine administration of antibiotics [5,6,7]. The evaluation of the treatment’s efficacy is complicated by the difficult differentiation of self-cure from cure rates and the lack of determination of the antibiotic sensitivity of bacteria present in uterine discharges [8].

Oxytetracycline and ceftiofur are among the most often used antibiotics in the treatment of metritis in cows [9]. According to the new regulations for the use of antibiotics in veterinary medicine, which were introduced in 2022, cefalosporins are among the restricted agents [10,11]. Oxytetracycline remains among the antibacterials with alternatives in human and veterinary medicine for their indications and with a low risk for the spread of resistance [10]. Therefore, the prudent use of antibiotics, such as oxytetracycline, based on a good knowledge of pharmacokinetics and contemporary information on the sensitivity of bacterial pathogens is essential for clinical efficacy [12].

Several articles have described the pharmacokinetics of oxytetracycline after its intrauterine administration and studied its disposition in blood and milk [13,14,15]. Increasing the severity of metritis, accompanied by a compromised barrier of the endometrium and a delay in involution, leads to a higher absorption of oxytetracycline from the uterus and results in higher milk concentrations of the drug, which leads to a withdrawal time issue [15]. According to other authors, the intrauterine absorption of oxytetracycline is poor but residues can be still of concern [16,17]. The available information about the oxytetracycline concentrations in the uterine tissues and uterine secretion is based on the intravenous and uterine administration of the antibiotic in healthy cows [18]. It does not reflect the changes in inflamed tissue, such as increased blood flow to the uterus, the breakdown of epithelial barriers and changes in the uterine fluids as well as the levels of inflammatory proteins, which may influence the disposition of oxytetracycline and the effective recovery of the endometrium [15,19]. Oxytetracycline administration can cause irritation to the endometrium and the infiltration of polymorphonuclear leukocytes in the uterine secretion, which can result in strain on the cows and the partial elimination of antibiotics from the uterine lumen [16,17]. Published data about the changes in the uterine secretion levels of the antibiotic are scarce. There is no information about the retention of oxytetracycline in the uterus and the maintained concentrations in the uterine secretion after the application of the drug by intrauterine infusion. Moreover, the levels of antibacterial drugs after intrauterine administration have not been discussed in relation to the sensitivity to pathogenic bacteria isolated from the uterine secretion.

The current study was designed because a widely accepted protocol for the treatment of metritis in cows is not yet available; the best strategy for oxytetracycline administration in the treatment of this puerperal disease is still not acknowledged and there is existing evidence that systemic oxytetracycline does not penetrate the uterine secretions at effective concentrations [20]. The aim was to evaluate the levels of oxytetracycline in the uterine secretion, as well as in plasma and milk, following the intrauterine administration of the drug for three consecutive days in cows with puerperal metritis. Additionally, to support the efficacy of local treatment with oxytetracycline, the isolation of pathogenic bacteria was performed before and after the treatment and their sensitivity to oxytetracycline was determined.

## 2. Materials and Methods

### 2.1. Drugs and Reagents

Oxytetracycline 10% solution (oxytetracycline hydrochloride in propylene glycol carrier, Alfasan International B.V., Woerden, The Netherlands) was used for the intrauterine treatment of the cows. The animals received the antibiotic by the intrauterine route of administration. Oxytetracycline hydrochloride ≥95% crystalline and doxycycline hyclate with a purity of ≥98% (HPLC grade, Sigma-Aldrich, St. Louis, MO, USA) were used as analytical standards. The following reagents were used for extraction and for the further analysis of the drugs: trifluoroacetic acid (99.5%) (Fisher Chemical, Fisher Scientific, Waltham, MA, USA), acetonitrile OPTIMA^®^, LC–MS grade (Fisher Chemical, Fisher Scientific, Waltham, MA, USA), methanol, LC–MS grade (CHROMASOLV LC–MS, Honeywell, Seelze, Germany), formic acid for mass spectrometry ~98% (Honeywell Fluka™, Seelze, Germany) and water for chromatography (LC–MS grade LiChrosolv^®^, Merck KGaA, Darmstadt, Germany).

### 2.2. Animals

The experiments were performed in the Training Experimental Farm of Trakia University, Stara Zagora, Bulgaria, between April 2020 and May 2021. The study included six lactating cows of different breeds and was designed according to the rules of Bulgarian legislation (Ordinance No. 20/1.11.2012 on the minimum requirements for protection and welfare of experimental animals and requirements for use, rearing and/or their delivery, License 151/26.09.2016). The diet was adjusted to the requirements of lactating dairy cows and water was supplied ad libitum. Milking was carried out twice daily (7:00 h and 17:00 h).

All the animals included in the study showed clinical signs of metritis: purulent discharge from the uterus. Healthy animals (a control group) were not subjected to intrauterine treatment with oxytetracycline because antibiotic administration in healthy animals and repetitive catheterization can cause disorders in the normal involution of the uterus, closure of the cervix and injuries of the cervix and dysbiosis, which can adversely impact reproductive outcomes [10]. Uterine health was determined by rectal examination on days 5, 10, 15 and 21 after parturition and after the checking of the uterine content. The diagnosis of metritis was confirmed by rectal examination after the appearance of abnormal vaginal discharge, which was registered by the vet on the farm. The system of Sheldon et al. [21] was applied to determine the grade of the clinical metritis. All animals were diagnosed with clinical metritis grade 1. Clinical signs included an abnormally enlarged uterus, a purulent uterine discharge without fever and any other systemic signs of illness. The cows did not show additional signs of systemic disorders, such as decreased milk yield, dullness, fever of >39.5 °C or toxemia, which are typical of grade 2 or grade 3 clinical metritis, respectively [21]. After a complete medical check, the health status of the cows was monitored every day by following the changes in feed intake, condition and udder filling. The animals did not show an increase in body temperature or the clinical signs of other diseases. At the end of the experiment, the cows underwent a medical check. They were considered healthy when the response to the treatment resulted in the absence of clinical signs of uterine infection and the recovery of ovarian function.

Detailed information about the animals included in the experiment is given in Table 1.

### 2.3. Experimental Design

Individual blood samples were obtained in heparinized tubes (2.5 mL lithium heparin, FL Medical, Italy) from the vena epigastrica cranialis superficialis before drug administration, on the day of treatment, from individual cows. Milk and uterine secretion samples, needed for microbiological examination, were aseptically obtained in sterile tubes (10 mL), at the same time. Catheterization with a sterile catheter was applied to obtain uterine secretion samples. After this, the cows were treated with the administration of oxytetracycline as a 10% solution (Alfasan International B.V., Woerden, The Netherlands) in the uterus. The animals were treated once daily for three consecutive days, at a dose rate of 8 mg/kg bw, between 8 and 8:30 h a.m. after the morning milking. All samples were immediately transported to the laboratories of the faculty. The microbiology lab received the milk and uterine secretion samples for microbiological tests. Milk samples from every cow were stored for pharmacokinetic studies and for the preparation of standard curves during the analysis of oxytetracycline concentrations. Blood samples were centrifuged at 1500× *g* for 10 min. Plasma from every cow included in the study was separated in clean tubes and was frozen at −25 °C. It was used for the preparation of standard curves for drug analysis. After the third uterine drug administration, blood and milk (10 mL) samples were obtained at the following time intervals: 0, 0.5, 1, 3, 6, 9, 12, 24, 48, 72 and 96 h. The obtained plasma was stored at −25 °C until analysis. Milk samples used for pharmacokinetic analysis corresponded to the time of complete milking and were obtained at 0, 0.5, 9, 24, 48, 72 and 96 h after the last treatment. A pooled sample from all four quarters was obtained from the collecting vessel after the complete milking of the cows, from every cow. At the other intervals, complete milking was not possible and pooled milk samples from the four quarters of every cow were used to compare the levels with those in the blood and in the uterus. Samples from uterine secretion were obtained via sterile catheter 6, 24, 30, 48, 54, 72, 78 and 96 h after oxytetracycline administration. Uterine secretion was obtained 10 and 17 days after the start of the treatment (Figure 1). All the samples were immediately stored at −25 °C until analysis.

### 2.4. Isolation and Identification of Pathogenic Bacteria and Determination of Minimum Inhibitory Concentration (MIC)

Samples from uterine secretion and from milk were seeded on tryptic soy agar (TSA, HiMedia, Thane West, India) and MacConkey agar (HiMedia, Thane West, India). They were incubated at aerobic conditions for 24–72 h. Aerobic mesophilic bacterial pathogens causing metritis in cows were isolated in a previous study on the same farm [20]. It was anticipated that we would find similar bacterial species. Based on these results, the isolation of pathogenic bacteria was also performed on TSA supplemented with 5% defibrinated sheep blood. The isolates were subjected to some specific tests for characterization after initial identification, as described by Mileva et al. [20]. The tests in the microbiology lab were performed according to the manufacturer’s instructions and the general rules for aseptic work [22] (p. 105). Microplates of generation GenIII were used for the phenotypic identification of the bacteria. They were run on a semi-automated system, BioLog (BioLog, Hayward, CA, USA). The conditions of the incubation of the plates were aerobic, at 33 °C, and the process lasted 20–24 h.

From the uterine samples obtained before oxytetracycline administration, *Trueperella pyogenes* was isolated from three cows, *Streptococcus intermedius* from one cow, *Escherichia coli* from one cow and *Bacillus pumilis* from one cow. The second set of samples obtained seven days after the end of the treatment confirmed the persistence of the same isolates in every cow. A further seven days later, the last samples from the uterine secretion were obtained and *T. pyogenes* was isolated from one cow (Cow No. 2). The MICs of the pathogenic isolates were determined using the micro-dilution method in cation-adjusted Mueller–Hinton broth (MHB), according to CLSI Guidelines [23]. MHB, supplemented with 2% (vol/vol) lysed horse blood [23], was used for tests with *Trueperella pyogenes*. The conditions of the incubation of the plates were as follows: 37 °C for 48 h in an atmosphere enriched with 5% CO2. The optical density was read spectrophotometrically at a 620 nm wavelength (Synergy LX Multi-Mode Microplate Reader, BioTek, Santa Clara, CA, USA). The experiments were repeated three times with 4 independent replications.

### 2.5. LC–MS/MS Analysis

LC–MS/MS was used for the determination of oxytetracycline concentrations in plasma, milk and uterine secretion [24,25,26]. A previously described procedure was used for the extraction of the tetracycline antibiotic from the plasma samples [27]. Doxycycline was used as an internal standard (IS, 10 µg/mL). IS (50 µL) was added to 500 µL plasma. Trifluoroacetic acid (TFA, 65 µL) was added to the samples and they were subjected to vortexing for 1 min and subsequent centrifugation at 10,800× *g* at 4 °C for 10 min. The supernatant was filtered through 0.22 µm syringe filters and 5 µL was injected into the LC–MS/MS system. The same procedure was used for the extraction of oxytetracycline from the uterine secretion. After the centrifugation step, the supernatant was purified using a 1 mL cartridge (Captiva EMR-Lipid, Agilent, Santa Clara, CA, USA). Thereafter, the filtrate from the uterine secretion samples was diluted to fit the concentrations within the range of the standard curve. Several dilutions were prepared, and the analysis started with the highest dilution. One sample, diluted at different degrees (1:10 dilution steps), was analyzed several times. It was impossible to obtain a sufficient amount of uterine secretion from non-treated cows; therefore, the calibration curve for plasma was used to determine the levels in this matrix. An aliquot of 1000 µL milk was mixed with 100 µL IS and 130 µL TFA by vortexing for 1 min. The samples were centrifuged for 10 min at 10,800× *g* at 4 °C. The supernatant was purified with the help of a 1 mL cartridge (Captiva EMR-Lipid, Agilent, CA, USA) and the filtrate was placed in LC–MS/MS vials.

The purified extracts were injected into the Agilent 6460C Triple Quadrupole LC–MS/MS system with an AJS ESI ionization source. It was coupled to a liquid chromatography (LC) unit consisting of a 1260 Infinity II quaternary pump and a 1260 Infinity II Vial Sampler. A Poroshell 120 EC C18 column (4.6 mm i.d. × 100 mm, 2.7 µm, Agilent Technologies, Santa Clara, CA, USA) was used for the separation of tetracycline antibiotics. A positive ion mode was applied for the determination of the studied antibiotics. Nitrogen was used as a drying gas at 12 L·min^−1^, as a nebulizer gas at 45 psi and a sheath gas at 400 °C and sheath flow 12 L·min^−1^. The gas temperature was 350 °C. The other parameters were as follows: capillary voltage: 4000 V; nozzle voltage: 500 V; and dwell time: 200 ms. Oxytetracycline was determined according to the qualifying ion 461.1 *m*/*z* and the quantifying ions 444.0 *m*/*z* and 443.1 *m*/*z*, respectively. The IS doxycycline was determined by qualifying ion 445.1 *m*/*z* and the quantifying ions 428.10 *m*/*z* and 410 *m*/*z*, respectively. Mobile phase A was prepared with LC–MS/MS water and formic acid (0.1%). Mobile phase B was acetonitrile. The flow rate was 0.3 L·min^−1^ and the run time was 12 min, with a post-run of 4.5 min. The gradient program was set as follows: 0–0.5 min (90% A, 10% B) and 0.5–8 min (2% A, 98% B). Mass Hunter Software (Agilent Technologies, Santa Clara, CA, USA) was used for the acquisition of the data and the quantitative analysis. The method was validated with a limit of detection (LOD) of 6.92 ng·mL^−1^ and limit of quantification (LOQ) of 20.98 ng·mL^−1^. The intra-day variability was 3.29% and the inter-day variability was 10.13%. The mean recovery was 97% and the mean accuracy was 95%. The standard curve for plasma was linear between 10 and 2000 ng·mL^−1^ (R^2^ = 0.9989) and that for milk was linear between 10 and 1000 ng·mL^−1^ (R^2^ = 0.9946). The LOD of 10 ng·mL^−1^ and LOQ of 30 ng·mL^−1^ were calculated for milk. The values of the mean accuracy and mean extraction recovery of oxytetracycline for milk were 99.53% and 88.86%. The mean intra- and inter-day variations for the same matrix were 4.63 and 8.70.

### 2.6. Pharmacokinetic Analysis

The oxytetracycline plasma concentration vs. time curve was obtained by non-compartmental analysis based on statistical moment theory using Phoenix 8.3.4 software (Certara^®^, Cary, NC, USA). The non-compartmental approach was applied for the analysis of data for plasma, milk and uterine secretion. The cut-off value for the goodness of fit was set at R^2^ > 0.80. The area under the curve (AUC) was calculated by the linear trapezoidal linear interpolation rule to the last quantifiable drug concentration–time point (C_t_) and infinity. Cut-off values for the percentage of extrapolation of AUC were set to <20%. Therefore, AUC was presented only for milk. AUC and area under the first moment curve (AUMC_0–∞_) were used for the calculation of the mean residence time (MRT_0–∞_) for milk. The data from the terminal phase of the log plasma or milk concentration versus time curve were used for the estimation of the elimination rate constant (k_el_). The mean maximum concentration (C_max_) and time to obtain the maximum concentration (T_max_) for plasma, milk and uterine secretion were calculated on the basis of the observed values.

### 2.7. Statistical Analysis

Pharmacokinetic parameters were presented as the geometric mean, with the minimum and maximum in brackets [28]. The statistical evaluation of the data was performed after testing for a normal distribution. Statistically significant differences in pharmacokinetic parameters between plasma, milk and uterine secretion were determined with the Mann–Whitney test (STATISTICA for Windows 10.0, StatSoft, Inc., Tulsa, OK, USA). A *p* value less than 0.05 was accepted as statistically significant. The concentrations in the uterine secretion were presented as median and range.

## 3. Results

### 3.1. Pharmacokinetics of Oxytetracycline

Oxytetracycline in plasma was found at low concentrations as early as the first sampling point, 0.5 h after drug administration, in five out of six cows (Appendix A). The measured values of the samples obtained between 0.5 and 48 h were close to the limit of quantification. The antibiotic concentrations were under the limit of quantification at the later sampling points. The plasma concentrations for cow No. 2 were below the LOQ in most of the samples (Appendix A). C_max_ in plasma was significantly lower (*p* < 0.05) in comparison to the maximum levels in milk and uterine secretion. The t_1/2el_ value was relatively high, although the plasma concentrations were low, and the drug was not found 48 h after the last oxytetracycline administration in four out of six cows (Table 2).

Oxytetracycline penetrated the milk and its concentrations were measured in the samples from all cows included in the experiment. In line with the observations for plasma samples, concentrations of the antibiotic in milk were found only at two sampling points in cow No. 2. Oxytetracycline was found in the milk at all sampling intervals for the rest of the animals (Appendix A). These levels were higher than the levels in plasma but lower in comparison to the concentrations in the uterine secretion (Table 3). The milk/plasma ratio of oxytetracycline concentrations showed that the levels of the antibiotic in milk remained higher than those in the plasma during the sampling period (Table 4). The median values of C_max_ were four-fold higher in comparison to those in plasma (*p* < 0.05). The ranges of values for T_max_ and for the elimination of the half-life were similar for the milk and the plasma (Table 2).

The intrauterine administration of oxytetracycline resulted in high values of the antibiotic in the uterine secretion (Table 3). Significant inter-individual variations in the levels of antibiotic were found. The uterine secretion C_max_ levels were statistically significantly higher in comparison to those in plasma and milk (Table 4). T_max_ values were 17.01 (6.0–78.0) h. Taking into account the route of oxytetracycline administration, the real T_max_ could be expected immediately after drug application in the uterus. Data from Table 3 show that oxytetracycline was found in the uterine secretion 96 h after the drug’s application at concentrations much higher than those in the plasma and in the milk.

### 3.2. MIC Concentration of Oxytetracycline against Pathogenic Isolates

Gram-positive and Gram-negative bacteria were isolated from the uterine secretion. *Trueperella pyogenes* was isolated from three cows, *Streptococcus intermedius* from one cow, *Escherichia coli* from one cow, as well as *Bacillus pumilis* (*n* = 1 cow). Pathogenic bacteria were not identified in the milk samples. A sensitivity test was performed, and minimum inhibitory concentrations were determined. The bacterial species often associated with clinical metritis in cows, *Trueperella pyogenes*, had MIC values between 16 (*n* = 2 isolates) and 64 µg·mL^−1^ (*n* = 1 isolate). An MIC value of 8 µg·mL^−1^ was determined for *Escherichia coli*. These values for *Streptococcus intermedius* and *Bacillus pumilis* were 4 µg·mL^−1^ and 8 µg·mL^−1^, respectively. These values were 2- to 6-fold lower than the concentrations in the uterine secretion. The *Trueperella pyogenes* isolate, with an MIC value of 64 µg·mL^−1^, was found in the uterine secretion samples from the second cow before and after the treatment.

At the end of the study, all the animals were clinically cured and showed signs of a normal estrus cycle.

## 4. Discussion

The presented investigation was performed in a small dairy farm in Bulgaria with common problems with reproductive disorders. It was planned after a similar experiment with cows with clinical signs of metritis, which were treated intramuscularly with long-acting oxytetracycline [20]. *Trueperella pyogenes* was isolated from the uterine secretion in both studies, but this pathogen was not isolated from milk [20]. Among isolates from the cows enrolled in this study, pathogenic bacteria, such as *Streptococcus intermedius*, *Escherichia coli* and *Bacillus pumilis*, were also identified. Other studies associated the puerperal metritis with the overgrowth of the same bacterial species, as well as with the anaerobic or facultative anaerobic microorganisms *Fusobacterium necrophorum* and *Proteus mirabilis* [1,8,29].

Aerobic and anaerobic bacteria, commonly found in postpartum intrauterine secretion, determine the use of broad-spectrum antibiotics, such as oxytetracycline, administered systemically or by intrauterine infusion [9,30,31]. A previous study revealed that the systemic administration of oxytetracycline as a long-acting drug formulation at a dose rate of 20 mg/kg did not guarantee the achievement of effective concentrations in the uterine secretions [20]. Therefore, it was important to evaluate the changes in the oxytetracycline concentrations in the uterine secretion with time and in relation to sensitivity of the pathogenic isolates after intrauterine administration at a dose rate of 8 mg/kg in cows with puerperal metritis.

There is evidence that after the intrauterine infusion of the studied tetracycline antibiotic, absorption is possible. Its concentrations in the uterine tissues are similar regardless of the route of administration [18,19]. The penetration of oxytetracycline in the uterine tissues is important for the outcome of the treatment because infections are not restricted to the uterine cavity. The intrauterine administration of oxytetracycline 10% solution for three consecutive days led to the absorption of the drug from the uterus and excretion through the milk. These results confirmed previously reported data about the distribution of the tetracycline antibiotic in body fluids after the same route of administration. Gorden et al. [19] reported two-fold higher plasma levels of oxytetracycline (220.6 ± 31.0 ng·mL^−1^) achieved later, at T_max_ of 23.2 ± 1.3 h, when compared to our data. Masera et al. [18] found similar levels in plasma as in the cited study, 270 ± 250 ng·mL^−1^, measured 12 h after oxytetracycline application in the uterus as a propylene glycol solution (50 mg·mL^−1^). Similarly to previous investigations, high between-subject variability was observed in the current study: in some cows, C_max_ values were 280 ng·mL^−1^, but in two animals, they were <100 ng·mL^−1^. Measurable levels of the antibiotic remained in circulation longer, until the last sampling point (96 h), in the cited study [18], versus 48 h in the current investigation. The differences can be attributed to the administration of a long-acting dosage form of oxytetracycline at a dose rate of approximately 6.7 mg/kg. Oxytetracycline in the form of pessaries, administered in the uterus of healthy cows at a dose rate of 3 g/cow (approximately 4.6 to 6 mg/kg), was absorbed to a higher degree, with a C_max_ of 550 ± 280 ng·mL^−1^ and an elimination half-life of 21.96 ± 4.42 h [13]. Altogether, these data indicate that oxytetracycline is absorbed after intrauterine administration and can be found in the central circulation, although at low concentrations.

The absorption of oxytetracycline from the uterus is also reflected by the achievement of measurable concentrations in the milk. The intrauterine administration of a single oxytetracycline dose of 8 mg/kg in healthy cows resulted in milk levels of 120–210 ng·mL^−1^, 12 and 24 h after treatment, respectively [18]. Our data showed that the levels of the antibiotic in the milk were higher in comparison to those in the plasma. Similar results were reported in a published investigation in cows with metritis indicating that higher concentrations can be expected in milk in cases with more severe inflammation of the uterus [19]. Published C_max_ mean values of oxytetracycline in milk were between 136.1 and 201 ng·mL^−1^, which were achieved at a T_max_ of 12 to 22 h after intrauterine infusion [19]. The higher concentrations found in our study, and the higher values of AUC 19.63 h·µg·mL^−1^ in comparison to the reported AUC of 5.96 h·µg·mL^−1^, can be explained by the differences in the applied drug formulations and the dosing regimens: 8 mg/kg for three consecutive days versus approximately 3.35 to 6.7 mg/kg as a single dose [19,32]. According to Gorden at al. [19], oxytetracycline residues in milk can persist in some cows for 96 h. Solid dosage forms as pessaries resulted in C_max_ of 209 ng·mL^−1^ in milk and the retention of drug residues for 72 h after oxytetracycline administration in healthy Friesian dairy cows [13]. Our data confirmed these observations regarding the presence of measurable antibiotic levels for a long period after the cessation of the drug administration, despite the differences in the applied dosing regimens and drug formulations. Veterinarians should acknowledge the withdrawal time after the intrauterine administration of oxytetracycline. The measured concentrations of oxytetracycline were lower than the MRL of 100 µg·L^−1^ 96 h after treatment in five of the cows, but, in one animal, it remained almost two-fold higher than the cut-off value. The severity of uterine inflammation can contribute to the higher secretion of oxytetracycline in the milk and to the prolongation of the withdrawal time due to increased blood flow during uterine inflammation, together with higher penetration through the blood–milk barrier [19,20,33]. Nevertheless, a shorter withdrawal time can be expected after the intrauterine administration of 10% oxytetracycline for three consecutive days if compared to a single intramuscular injection of a long-acting drug formulation containing 20% of the antibiotic [20].

Successful antibacterial treatment requires an understanding of the pharmacokinetics of the antibiotics together with their pharmacodynamics and the acknowledgement of the characteristics of the host, pathogen and drug. Therefore, it is essential to obtain information about the retention of oxytetracycline in the uterine secretion and thus in the uterus. There are some investigations about the penetration of oxytetracycline in the uterine secretion after its systemic administration, and although higher levels could be expected in cows with metritis, they were below the effective concentrations [18,20]. The local application of oxytetracycline expectedly resulted in very high concentrations in the uterine secretion. The first sampling time in the current study was six hours after the uterine infusion of the antibiotic, and, in two cows, C_max_ was >1.5 mg·mL^−1^. The value of C_max_ of 471.15 µg·mL^−1^, within a range of 176.13–1668.76 µg·mL^−1^, reflected high inter-individual variations between the cows. The individual sensitivity of the endometrium to oxytetracycline, and the different breeds, age and periods for the development of postpartum clinical metritis, can contribute to the explanation of the observed variations [34]. Only one study measured the concentrations of oxytetracycline in the uterine secretion after local application. Masera et al. [18] did not determine the exact oxytetracycline concentrations in the uterine secretion, but levels > 4 µg·mL^−1^ were found 24 h after the intrauterine administration of a dose of 4 mg/kg. The lower concentrations in the uterine secretions of cow No. 2, in comparison to all other animals, deserve attention. This cow was older, and the metritis was diagnosed a week later in comparison to the other cows in the study. All the other animals showed fluctuations in oxytetracycline concentrations around 100 µg·mL^−1^ 72 h after the treatment. These high concentrations serve as a deposit from which the antibiotic can be absorbed by the uterine tissues in order to contribute to treatment efficacy. The absence of the exact data in the published literature does not allow the comparison of the results, but the plasma and milk concentrations were far below the uterine secretion levels.

The achieved oxytetracycline levels in the uterine secretion are a prerequisite for a successful decrease in the bacterial load or even eradication of the pathogenic bacteria. The obtained concentrations were equal to or higher than MIC values ≤ 64 µg·mL^−1^, and they were retained in the uterine secretion most of the time in five of the cows. The pathogenic bacteria were re-isolated 10 days after the start of the treatment but were not found 17 days after the first day of oxytetracycline infusion. *Trueperella pyogenes* was re-isolated at the last sampling point from cow No. 2, which can be explained by the very low oxytetracycline levels (≤3 µg·mL^−1^) maintained in the uterine secretion of this animal. As reported previously, the re-isolation of this pathogen was found in cows when low concentrations of the antibiotic were measured in the uterine secretion [20]. The MIC values of *Trueperella pyogenes* were within the reported concentrations between 0.25 and ≥128 µg·mL^−1^ [35,36]. *Trueperella pyogenes* isolates from the farm at which the investigation was carried out showed no potential for biofilm formation [37]. *Trueperella pyogenes* is very often isolated from uterine secretions obtained from cows with metritis [1,3]. The use of oxytetracycline is recommended in such cases. It was associated with a higher first service conception rate in groups treated with this drug if compared to other antibiotics [38,39]. Although, so far, there is no established clinical cut-off value or epidemiological cut-off value for this pathogen, our data demonstrated that effective oxytetracycline levels can be maintained for at least 78 h and, in some cows, for 96 h after the last application of the antibiotic. Clinical efficacy in terms of the alleviation of inflammation and the restoration of the estrus cycle was observed in all animals in our study, which can be related to the sufficiently high antibiotic levels at the site of action. This clinical outcome differs from the results reported in a previous experiment, where cows with metritis were treated intramuscularly with oxytetracycline [20]. In the cited study, the inflammation of the uterus became chronic in two cows and 10% povidone iodine solution was applied as an additional treatment. The treatment failure of metritis was explained by the limited penetration of oxytetracycline into the uterus and higher MIC values than the concentrations of the antibiotic. The comparison of both studies in cows with metritis, performed in the same farm and within close periods of time, suggested that the treatment of the uterine inflammation was more readily observed after infusion of oxytetracycline in the uterus than after intramuscular injection of a long-acting dosage form. In line with other investigations, this proves that bacteriological tests are necessary for selective antibiotic treatment, leading to a reduction in the application of antibiotics [40]. The longer period needed for the restoration of the estrus cycle and uterus in one cow can be explained by multiple factors, including the age, the start of the treatment and the changes in uterine microbiota after application of the broad-spectrum oxytetracycline [2].

## 5. Conclusions

The limited number of animals included in our investigation is a limitation that does not allow us to offer a general and strong conclusion about the efficacy of oxytetracycline in the treatment of cows with metritis. The oxytetracycline concentrations in the uterine secretion were higher than the MIC values of the isolated pathogenic bacteria during the treatment. The intrauterine infusion of oxytetracycline for three consecutive days resulted in the successful treatment of the acute inflammation of the uterus known as metritis and the restoration of the estrus cycle. Oxytetracycline was absorbed from the uterus and reached higher concentrations in milk than MRL values, which indicates the need to monitor the withdrawal time in order to ensure consumer safety.

## Figures and Tables

**Figure 1 animals-12-01922-f001:**
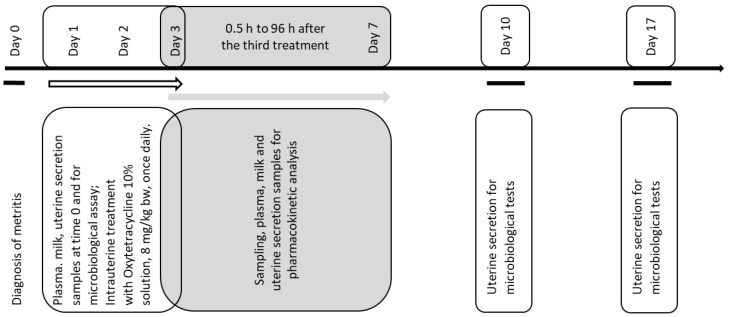
Experimental design of the study with timeline of the treatments and sampling.

**Table 1 animals-12-01922-t001:** Information about cows included in the investigation.

Cow No.	Breed	Body Weight (kg)	Age (Years)	Days after Parturition	Mean Daily Milk Yield (L)
1	Bulgarian black and white cattle	585	6	10	14
2	Bulgarian black and white cattle	545	11	17	20
3	Jersey cattle	355	6.5	11	12
4	Bulgarian black and white cattle	531	2.5	11	11
5	Simmental cattle	566	2.5	10	23
6	Simmental cattle	543	2.5	9	24.5

**Table 2 animals-12-01922-t002:** Pharmacokinetic parameters of oxytetracycline after intrauterine administration in cows (*n* = 6) at a dose rate of 8 mg/kg oxytetracycline as 10% solution for three consecutive days.

Parameters	Units	Geometric Mean (Minimum–Maximum)
Non-compartmental analysis—plasma		
k_el_	h^−1^	0.021 (0.016–0.029)
t_1/2el_	h	32.60 (24.12–44.23)
T_max_	h	1.28 (0.5–9.0)
C_max_	µg·mL^−1^	0.11 (0.06–0.28) ^a,b,c^
AUC_0−t_	h·µg·mL^−1^	1.72 (1.12–2.48)
Non-compartmental analysis—milk		
k_el_	h^−1^	0.024 (0.02–0.034)
t_1/2el_	h	28.15 (20.31–34.81)
T_max_	h	5.05 (0.5–9.0)
C_max_	µg·mL^−1^	0.39 (0.23–1.18) ^a^
AUC_0−t_	h·µg·mL^−1^	19.63 (10.96–57.47)
AUC_0–∞_	h·µg·mL^−1^	22.20 (12.82–65.74)
Extrapolation of AUC	%	11.27 (7.64–14.54)
AUMC_0–∞_	h^2^·µg·mL^−1^	1015.42 (596.50–3032.01)
MRT	h	45.74 (43.39–47.99)
Non-compartmental analysis—uterine secretion		
C_max_	µg·mL^−1^	471.15 (176.13–1668.76) ^b,c^

T_max_: time of C_max_; C_max_: maximum concentration; k_el_: elimination rate constant; t_1/2el_: elimination half-life, which is presented as harmonic mean; AUC_0–t_: area under the concentration–time curves from 0 to last sampling time t; AUC_0–∞_: area under the concentration–time curves from 0 to infinity ∞; AUMC_0–∞_: area under the first moment curve; MRT: mean residence time; ^a^: statistically significant differences between parameters for plasma and for milk; ^b^: statistically significant differences between parameters for plasma and for uterine secretion; ^c^: statistically significant differences between parameters for milk and for uterine secretion.

**Table 3 animals-12-01922-t003:** Measured individual concentrations of oxytetracycline in uterine secretion (*n* = 6) after the third intrauterine dose (8 mg/kg) of 10% solution of oxytetracycline.

Time (h)	Cow 1 (µg·mL^−1^)	Cow 2 (µg·mL^−1^)	Cow 3 (µg·mL^−1^)	Cow 4 (µg·mL^−1^)	Cow 5 (µg·mL^−1^)	Cow 6 (µg·mL^−1^)	Median (Min–Max)
6	1632.34	433.79	52.87	39.17	1668.76	nd	433.79 (39.17–1668.76)
24	15.84	3.10	159.34	18.16	16.88	19.02	17.52 (3.10–159.34)
30	38.93	3.12	157.41	187.15	193.29	251.90	172.28 (3.12–251.90)
48	43.46	nd	176.13	165.49	184.56	151.08	165.49 (43.46–184.56)
54	134.12	2.14	156.50	nd	178.28	171.01	156.50 (2.14–178.27)
72	87.67	1.51	nd	151.66	166.97	183.97	151.66 (1.51–183.97)
78	58.54	1.22	nd	208.64	184.81	236.95	184.81 (1.21–236.95)
96	5.32	1.59	17.58	156.85	467.55	151.08	84.33 (1.58–467.55)

nd—not determined.

**Table 4 animals-12-01922-t004:** Milk/plasma ratio, uterine secretion/plasma ratio and uterine secretion/milk ratio (median and range min–max) of oxytetracycline concentrations in cows (*n* = 6) with clinical metritis after the intrauterine administration of 8 mg/kg of 10% solution of oxytetracycline for three consecutive days.

Time (h)	Milk/Plasma Ratio	Uterine Secretion/Plasma Ratio	Uterine Secretion/Milk Ratio
0.5	3.67 (1.67–12.35)	-	
1	2.53 (0.51–9.45)	-	
3	4.08 (2.80–14.31)	-	
6	7.82 (5.38–13.67)	10,341.13 (591.25–41,806.29)	1887.32 (55.05–6587.55)
9	6.81 (2.28–14.71)	-	-
12	9.63 (3.81–24.10)	-	-
24	6.14 (3.57–19.25)	515.41 (258.77–3488.97)	117.72 (43.22–181.28)
48	13.33 (6.00–19.62)	6205.34 (3305.75–7151.77)	629.85 (247.90–2881.18)
72	-	-	1946.27 (938.40–2804.94)
96	-	-	2887.82 (99.67–6492.52)

## Data Availability

The data presented in this study are available in the Appendix A of this manuscript.

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
