# Peer review of "Oxytetracycline Persistence in Uterine Secretion after Intrauterine Administration in Cows with Metritis"

_animals, 2022, doi:10.3390/ani12151922_

Round 1

Reviewer 1 Report

Interesting study, but as stated by the Authors with a main weakness regarding the number o f animals.

I would expect all cows bred, this lately increase the aforementioned weakness

Author Response

Reviewer 1

Interesting study, but as stated by the Authors with a main weakness regarding the number of animals.

I would expect all cows bred, this lately increase the aforementioned weakness.

Answer:

Thank you to the comments. We agree that from clinical point of view the number of the animals is low and control group is extremely important. However, our aim was to evaluate the changes in the concentrations of oxytetracycline in the uterine secretion and the persistence of the antibiotic in the uterine cavity. Usually pharmacokinetic studies are performed with limited number of animals and these data can serve for further population studies. With the help of all reviewer remarks, we hope that the manuscript was improved.

Reviewer 2 Report

In this study, the authors report the findings regarding the transfer and persistence of Oxytetracycline from the uterus of cows with clinical metritis towards the blood and milk and evaluate the duration of its persistence, as measured by LC-MS/MS analysis.

The study has potential interest for the cattle practitioner. Yet the low number of cows enrolled in the study together with the inexistence of a control group (postpartum cows without clinical metritis) may raise some concerns about the soundness of the study.

Furthermore, the authors should better describe the population, the moment of the diagnosis for clinical endometritis, the inclusion criteria for the diagnosis, the relative amount of fluids in the uteri of cows (it would vary according to time postpartum at treatment), and information related to the treatment implementation (the formulation used for Oxytetra - I would suspect liquid from the tables headings, but it can be an oil or aqueous solution-, with or without previous uterine lavage, etc). could the amount of uterine content impact the intra-uterine distribution of the Oxytetracycline given locally? it is important to determine a putative influence between these parameters and the concentration of the antibiotics in uterine fluids. Also, the authors should define the criteria used to establish the response to treatment: was it the clearance of the uterine (macro-)infection? the recovery of ovarian function?

I strongly recommend the introduction of an image depicting the moment for treatment and the diverse sample collection to clarify the study design.

Some additional minor suggestions or comments were introduced in the commented copy of the MS uploaded with this report

Author Response

Reviewer 2

In this study, the authors report the findings regarding the transfer and persistence of Oxytetracycline from the uterus of cows with clinical metritis towards the blood and milk and evaluate the duration of its persistence, as measured by LC-MS/MS analysis.

The study has potential interest for the cattle practitioner. Yet the low number of cows enrolled in the study together with the inexistence of a control group (postpartum cows without clinical metritis) may raise some concerns about the soundness of the study.

 Answer:

The study aimed to follow the changes of oxytetracycline concentrations in intrauterine secretion of cows with metritis. The majority of the published papers related to pharmacokinetic studies are performed with six animals and this number is widely accepted. First, pharmacokinetic investigations include intense sampling from one animal and second, many samples are generated in a single study which increases significantly the price for sample preparation and analysis. In such cases population pharmacokinetics helps a lot but design of population studies requires at least one conventional pharmacokinetic investigation. Absence of available literature about the disposition of oxytetracycline after its intrauterine administration provoke us to plan a full pharmacokinetic experiment with intense sampling. A published study with 3 healthy cows and 3 cows with metritis determined tissue levels of this tetracycline antibiotic (Bretzlaff et al. Am J Vet Res. 1983 May;44(5):764-9. PMID: 6869980). The animals were slaughtered at the end of the experiment in the cited study. It is almost impossible to obtain a permission for such an experiment. The published literature does not contain information for the persistence of oxytetracycline in the uterine secretion (Bretzlaff et al. Am J Vet Res. 1983 May;44(5):764-9. PMID: 6869980 and Bretzlaff et al., Am J Vet Res. 1983 May;44(5):760-3. PMID: 6869979).

Control group for intrauterine administration of oxytetracycline was not included because antibiotics should not be applied in healthy animals and such a treatment, without any inflammation, can cause more problems. Moreover, repetitive catheterization of the cervix for intrauterine application of drugs in healthy animals, especially during the post-partum period, can provoke disorders in the normal uterine involution and injuries of the cervix. We assumed that oxytetracycline will be used in clinical practice when inflammation exists and therefore we performed this study in cows with metritis and we did not wish to investigate oxytetracycline disposition in healthy cows (there are more than 20 oxytetracycline pharmacokinetic papers in healthy cows: https://pubmed.ncbi.nlm.nih.gov/?term=oxytetracycline%20pharmacokinetics%20in%20cows).

Furthermore, the authors should better describe the population, the moment of the diagnosis for clinical endometritis, the inclusion criteria for the diagnosis, the relative amount of fluids in the uteri of cows (it would vary according to time postpartum at treatment), and information related to the treatment implementation (the formulation used for Oxytetra - I would suspect liquid from the tables headings, but it can be an oil or aqueous solution-, with or without previous uterine lavage, etc). could the amount of uterine content impact the intra-uterine distribution of the Oxytetracycline given locally? it is important to determine a putative influence between these parameters and the concentration of the antibiotics in uterine fluids. Also, the authors should define the criteria used to establish the response to treatment: was it the clearance of the uterine (macro-)infection? the recovery of ovarian function?

 Answer: The information for the formulation was included at lines 98-99: “Oxytetracycline 10% solution (oxytetracycline hydrochloride in propylene-glycol carrier, Alfasan International B.V., The Netherlands)”

The treatment was performed without preliminary uterine lavage with any solutions but the uterus was subjected to massage for obtaining of the samples before application of oxytetracycline. After this procedure there was a negligible amount of secretion. The information about the population, the moment of the diagnosis for clinical metritis, the inclusion criteria for the diagnosis and information for the drug formulations were included in the manuscript (Table 1 and lines 120-138).

Criteria for the response to the treatment were included (lines 138-140).

I strongly recommend the introduction of an image depicting the moment for treatment and the diverse sample collection to clarify the study design.

Answer: A figure is included in the M & M.

Some additional minor suggestions or comments were introduced in the commented copy of the MS uploaded with this report

Answer: We tried to reflect all the remarks and to clarify the text.

Reviewer 3 Report

Major issues:

- The number of animals and their dispersion could be a negative point of the experimental design. Just 6 animals, three different breeds and very dispersed ages... In results, authors showed the importance of individual response on the ab concentrations on uterine fluid. Therefore I think that the sample size should be higher. As well, a more standardized herd or a cohort study could be planified. I encourage authors to increase sample size prior publishing this manuscript. 

Some other comments:

- ln 25-29. please rephrase, it is the same than in simple summary

- ln 47. italic? Check journal guidelines

- Introduction: I congratulate authors because of this section. It provides a good state-of-the-art. It would be welcome a short paragraph about "new approaches or insights" about new/different treatments than antimicrobials.

- ln 98. I recommend to say mg/kg, it is easier. Change it throughout the manuscript.

- ln 105. The number of animals and their dispersion could be a negative point of the experimental design. Just 6 animals, three different breeds and very dispersed ages... 

- ln 199. please explain a bit more about the metritis grade in order to enhance clarity for the reader.

- ln 130. mL. supervise the whole manuscript.

- ln 160. delete p.105, just include it it bibliography section.

- M&M. Please describe briefly how you took milk samples.

- ln 248. I find important to describe the animals used. They are just 6 of different ages and different breeds, thus, the table they included in suppl material, may be should be here. Or at least written in the manuscript.

- ln 256. Did you already describe this acronym?

- table 1. why did you provide a mean and range instead of mean +-sd? In table 2 you said median... Is it a no normal distribution?

- table 2. we appreciate the important individual factor on eliminating the ab in the uterine fluid. Is the sample size enough?

- table 3. I would like to see the changes in blood and milk, as showed for uterine fluis (table 2), instead of the ratios. Maybe you can summarize it all in one table like table 2, showing the results of uterine fluis, blood and milk of each animal.

- ln 303. I found no relevant this section. It would be very welcome to combine reproductive outcome with this study when comparing a healthy control group vs a metritis treated one, but, as this is not the aim of the study, I would delete this section.

- discussion: I would shorten this section. It is too long. Please start it resembling your main results. I would also compare the results with obtained by other authors and with another metritis degrees, as well, I would include advatages or disadvantages of using another treatments different than ab. 

Author Response

Reviewer 3

Thanks to reviewer for the critical remarks. We tried to fulfill all the requirements and to improve the presentation of the data.

Major issues:

- The number of animals and their dispersion could be a negative point of the experimental design. Just 6 animals, three different breeds and very dispersed ages... In results, authors showed the importance of individual response on the ab concentrations on uterine fluid. Therefore I think that the sample size should be higher. As well, a more standardized herd or a cohort study could be planified. I encourage authors to increase sample size prior publishing this manuscript.

Answer:

Usually, in a pharmacokinetic study no more than six animals are included in a group. The reason is intense sampling design. The sampling in the current investigation required more precise work due to receiving of samples from uterine secretion.

We completely agree that the animals were from different breeds but the farms in Bulgaria are not uniform in their breed in most cases. The permission to enter and to organize such an experiment in a big farm (for Bulgaria this means > 1000 cows) is extremely difficult. The Experimental farm of the University is under supervision of the Department of Obstetrics for years. During the examination period the rate of postpartum uterine diseases was 0.2% due to individual cases of dystocia and retained fetal membranes. These disorders did not show any breed specific appearance in the investigated farm. This farm has the highest conception rate after the second artificial insemination in cows without any postpartum disorders.

Some other comments:

- ln 25-29. please rephrase, it is the same than in simple summary

Answer: Revised.

 - ln 47. italic? Check journal guidelines

Answer: Revised.

 - Introduction: I congratulate authors because of this section. It provides a good state-of-the-art. It would be welcome a short paragraph about "new approaches or insights" about new/different treatments than antimicrobials.

Answer: There are some papers about the possibility to treat the endometritis in mares with N-acetylcysteine. There are papers which describes non-antibiotic treatment of endometritis in cows. These papers, however are not very closely related to the treatment of metritis in cows and we prefer to avoid this discussion in the current manuscript. All the described alternative methods that were used from clinicians in cattle practice are related to clinical or subclinical endometritis. We hope that this will be accepted by the Reviewer.

 - ln 98. I recommend to say mg/kg, it is easier. Change it throughout the manuscript.

Answer: Revised.

- ln 105. The number of animals and their dispersion could be a negative point of the experimental design. Just 6 animals, three different breeds and very dispersed ages... 

Answer:

An explanation was written above.

 - ln 199. please explain a bit more about the metritis grade in order to enhance clarity for the reader.

Answer: Performed.

 - ln 130. mL. supervise the whole manuscript.

Answer: Thank you for this remark. Done.

 - ln 160. delete p.105, just include it it bibliography section.

Answer: We followed the instructions for authors: “ For embedded citations in the text with pagination, use both parentheses and brackets to indicate the reference number and page numbers; for example [5] (p. 10). or [6] (pp. 101–105).”

 - M&M. Please describe briefly how you took milk samples.

Answer: Performed.

 - ln 248. I find important to describe the animals used. They are just 6 of different ages and different breeds, thus, the table they included in suppl material, may be should be here. Or at least written in the manuscript.

Answer: Performed, the table was placed in the manuscript.

 - ln 256. Did you already describe this acronym?

Answer: Done, in M & M.

 - table 1. why did you provide a mean and range instead of mean +-sd? In table 2 you said median... Is it a no normal distribution?

Answer: There are scientific papers according to which geometric mean (min-max) best describes the observed data, including the variability, in pharmacokinetic studies (Steven et al., Journal of Biopharmaceutical Statistics, 2000, 10:1, 55-71, DOI: 10.1081/BIP-100101013; Martinez et al., Pharmaceutics 2017, 9, 14. doi:10.3390/pharmaceutics9020014). The choice for the way of the presentation of the data was done after a test for normal distribution and according to the cited papers.

- table 2. we appreciate the important individual factor on eliminating the ab in the uterine fluid. Is the sample size enough?

Answer: The sampling intervals for uterine secretion were as many as possible and were planned after check of the available literature and the published information from similar investigations.

- table 3. I would like to see the changes in blood and milk, as showed for uterine fluis (table 2), instead of the ratios. Maybe you can summarize it all in one table like table 2, showing the results of uterine fluis, blood and milk of each animal.

Answer: This information was included as Table S2 in the supplementary file and now it is Table S1 in the same file.

- ln 303. I found no relevant this section. It would be very welcome to combine reproductive outcome with this study when comparing a healthy control group vs a metritis treated one, but, as this is not the aim of the study, I would delete this section.

Answer: The section was deleted.

 - discussion: I would shorten this section. It is too long. Please start it resembling your main results. I would also compare the results with obtained by other authors and with another metritis degrees, as well, I would include advatages or disadvantages of using another treatments different than ab. 

Answer: An explanation is given above. We tried to revise the discussion by omission of some results.

Reviewer 4 Report

Procedures

Subsection 2.3.

How may times per day was the drug administered, please clarify.

2.4.

Please present the details of identification of Streptococcus intermedius, as it might have been misidentified, due to the difficulties in typing.

I am extremely concerned due to the lack of control animals in this experiment.

Before I go any further in the evaluation of the manuscript, the authors MUST discuss this very significant omission.

Author Response

Reviewer 4

 Procedures Subsection 2.3. How may times per day was the drug administered, please clarify.

Answer:

A figure with precise indications of the days of treatment and the times per day is presented (once daily).

Information in the text is included: “The animals were treated once daily for three consecutive days, at a dose rate of 8 mg/kg bw, between 8 and 8:30 h a.m. after the morning milking.” (lines 149-150)

2.4.

Please present the details of identification of Streptococcus intermedius, as it might have been misidentified, due to the difficulties in typing.

Answer:

As described in the manuscript: „Identification of the bacteria from their phenotypic pattern was run on microplates of generation GenIII with the help of a semi-automated system (BioLog, USA). The plates were incubated under aerobic conditions at 33⁰C for 20–24 h.“ (lines 193-195)

The authors are aware that the phenotypic identification was not done with the most contemporary BioLog system but all the instructions for work with the system were followed and this system can give reliable results with certain limitations (Wragg et al., Journal of Microbiological Methods, 105, 2014, 16-21). We have experience with the work with this system in the identification of Streptococcus spp. of veterinary interest (Str. uberis and others, including Streptococcus intermedius). Unfortunately, we do not have access to MALDI-TOF MS for more precise identification.

I am extremely concerned due to the lack of control animals in this experiment.

Before I go any further in the evaluation of the manuscript, the authors MUST discuss this very significant omission.

Answer:

The aim of the study was to evaluate the changes of oxytetracycline levels in the uterine secretion with time, after intrauterine administration of the antibiotic.

Healthy animals (a control group) were not subjected to intrauterine treatment with oxytetracycline because antibiotic administration in healthy animals is contraindicated and can cause disorders in normal involution of the uterus, closure of the cervix and dysbiosis which can impact reproductive outcomes (included in the text at lines 122-125). We assumed that oxytetracycline will be used in clinical practice when inflammation exists and therefore we performed this study in cows with metritis and we did not wish to investigate oxytetracycline disposition in healthy cows (there are more than 20 oxytetracycline pharmacokinetic papers in healthy cows: https://pubmed.ncbi.nlm.nih.gov/?term=oxytetracycline%20pharmacokinetics%20in%20cows ).

A published study with 3 healthy cow and 3 cows with metritis determined tissue levels of tetracycline antibiotic (Bretzlaff et al. Am J Vet Res. 1983 May;44(5):764-9. PMID: 6869980). The animals were slaughtered at the end of the experiment in the cited study. It is almost impossible to obtain a permission for such an experiment nowadays. The published literature does not contain information for the persistence of oxytetracycline in the uterine secretion (Bretzlaff et al. Am J Vet Res. 1983 May;44(5):764-9. PMID: 6869980 and Bretzlaff et al., Am J Vet Res. 1983 May;44(5):760-3. PMID: 6869979). In a similar pharmacokinetic study (Gorden et al. J Dairy Sci 2016, 99, 8314–8322) healthy controls were not included. More animals were enrolled in the cited study in comparison to our investigation but samples from uterine secretion were not obtained and were not analysed. Blood sampling was less intense in comparison to our study.

Another problem is receiving of uterine secretion from healthy animals with relatively intense sampling time.

The text of the manuscript was additionally revised and it was focussed mainly to the data for the disposition of oxytetracycline after its intrauterine administration.

We hope that these explanations can be accepted as sufficient.

Round 2

Reviewer 3 Report

Authors accomplished all the issues I suggested, or even answer why they would rather to show it this way. 

I still thinking that the age distribution of cows could disturb the results, as well the low sample size and different breeds. A higher sample size and a more standardized design would be better.  

Author Response

Reviewer 3

Authors accomplished all the issues I suggested, or even answer why they would rather to show it this way. 

I still thinking that the age distribution of cows could disturb the results, as well the low sample size and different breeds. A higher sample size and a more standardized design would be better.  

Answer: We are thankful for the remarks. Unfortunately, at this stage we cannot add more patients. For very reach sample size we will need few years more looking at the intensity of appearance of cows with metritis in a relatively small farm. 

We hope that we will be able to collect more samples for a population PK study where the age and other factors can be introduced as co-variables and to test their impact on the PK parameters.

Reviewer 4 Report

Please improve the language of the manuscript before final acceptance.

Author Response

Reviewer 4

Please improve the language of the manuscript before final acceptance.

Answer: The manuscript has been revised by MDPI Editor. The certificate was attached.
